# Intestinal Behcet’s Disease: A Review of the Immune Mechanism and Present and Potential Biological Agents

**DOI:** 10.3390/ijms24098176

**Published:** 2023-05-03

**Authors:** Kun He, Xiaxiao Yan, Dong Wu

**Affiliations:** 1Department of Gastroenterology, Peking Union Medical College Hospital, Chinese Academy of Medical Sciences and Peking Union Medical College, Beijing 100730, China; 2Eight-year Medical Doctor Program, Chinese Academy of Medical Sciences and Peking Union Medical College, Beijing 100730, China; 3Clinical Epidemiology Unit, Peking Union Medical College Hospital, Chinese Academy of Medical Sciences and Peking Union Medical College, Beijing 100730, China

**Keywords:** intestinal Behcet’s disease, inflammatory bowel disease, immune mechanism, biological agents

## Abstract

Behcet’s disease (BD) is a chronic and recurrent systemic vasculitis involving almost all organs and tissues. Intestinal BD is defined as BD with predominant gastrointestinal involvement, presenting severe complications such as massive gastrointestinal hemorrhage, perforation, and obstruction in some cases. To some extent, intestinal BD is classified as a member of inflammatory bowel disease (IBD), as it has a lot in common with classical IBD including Crohn’s disease (CD) and ulcerative colitis (UC). Certainly, the underlying pathogenesis is not the same and dysregulation of immune function is believed to be one of the main pathogeneses in intestinal BD, although the etiology has not been clear up to now. Biological agents are an emerging category of pharmaceuticals for various diseases, including inflammatory diseases and cancers, in recent decades. Based on the deep understanding of the immune mechanism of intestinal BD, biological agents targeting potential pathogenic cells, cytokines and pathways are optimized options. Recently, the adoption of biological agents such as anti-tumor necrosis factor agents has allowed for the effective treatment of patients with refractory intestinal BD who show poor response to conventional medications and are faced with the risk of surgical treatment. In this review, we have tried to summarize the immune mechanism and present potential biological agents of intestinal BD.

## 1. Introduction

Behcet’s disease (BD) is a chronic and recurrent systemic variant vasculitis that mainly involves the skin, mucosa, joints, eyes, arteries and veins, nervous system and gastrointestinal organs [1,2]. Guan et al. in 2022 suggested that BD should be classified into eight subtypes: mucocutaneous BD, ocular BD, intestinal BD, cardiac BD, vascular BD, nervous BD, blood BD and articular BD [3]. Different subtypes present various disease courses, treatment responses and prognoses, suggesting a clinical heterogeneity and potential diversity of pathogenesis [4,5]. BD with gastrointestinal involvement is termed intestinal Behcet’s disease (intestinal BD). Intestinal BD can involve the entire gastrointestinal tract, most commonly the ileocecal region. Clinical manifestations include abdominal pain, diarrhea, nausea, vomiting, gastrointestinal bleeding, and perforation. A colonoscopy reveals ulcers and histopathology shows neutrophil infiltration in the vessel wall, perivascular and intravascular areas [6]. Intestinal BD can cause serious complications and has high rates of disability and mortality, showing poor outcomes and calling for active treatment [7]. The incidence of gastrointestinal BD is about 3–60% and it shows regional differences, with approximately 2.8–4.0% in Turkey, India and Saudi Arabia, 10% in China, 38–53% in Japan, and 50–60% in British [7]. According to a 30-year retrospective analysis in Japan, the clinical cluster of gastrointestinal involvement shows an increasing trend [8]. The diagnosis of intestinal BD mainly depends on clinical and endoscopic manifestations. It is worth noting that the relationship between intestinal BD and classical inflammatory bowel disease (IBD) including Crohn’s disease (CD) and ulcerative colitis (UC) is still disputed up to now [6]. Considering the similarity of genetic backgrounds, clinical manifestations, and therapeutic strategies between intestinal BD and classical IBD (especially CD), some experts place these two diseases in the same category or different spectrums of the same disease [9]. Certainly, there are still independent characteristics between intestinal BD and CD by careful clinical evaluation (Table 1), suggesting different underlying pathogenesis [10].

However, the pathogenesis of BD, including intestinal BD, is not clear. The well-known hypothesis reveals that the intricate interplay between environmental factors and genetic susceptibility causes immune dysfunction, which is thought to be an inflammatory disorder between autoimmune and autoinflammatory conditions [11]. The main conventional medications for intestinal BD include 5-aminosalicylic acids (5-ASAs), corticosteroids, and immunomodulators, although there are still several patients who are refractory to conventional treatment. For intestinal BD, mucosal healing is associated with a decreased risk of recurrence and surgery, and should be considered the outcome of monitoring disease activity and targets of management [12]. In the past decade, the adoption of biological agents in BD allowed for effective treatment of these refractory patients [5,13,14]. As 2018 European League Against Rheumatism (EULAR) recommendations suggested, for severe and/or refractory intestinal BD patients, monoclonal anti-tumor necrosis factor (TNF) antibodies should be considered [5]. A recent study reported that earlier and more aggressive treatment may be beneficial to patients with BD who had the highest risk for a severe disease course [15]. Based on the deep understanding of the immune mechanism of intestinal BD, biological agents targeting potential pathogenic cells, cytokines and pathways are optimized options [14,16]. In this review, we summarize the immune mechanism and present and potential biological agents of intestinal BD, aiming to improve clinicians’ understanding.

## 2. Immunity Mechanism

Both innate immunity and adaptive immunity show abnormalities in the pathogenesis of BD, which is activated by environmental factors in genetically predisposed patients [17,18,19]. For intestinal BD, abnormal activation of innate and adaptive immunity contributes to recurrent intestinal inflammatory reactions and destruction of intestinal vessels, the intestinal barrier and a series of injuries by inducing changes in the complex cytokine network and pathways [20].

### 2.1. Immunogens

Immune reactions are significantly important in the pathogenesis of BD, and infections, as one of various environmental factors, have been proven to be an important triggering factor. Bacterial infections, such as *Streptococcus sanguis* and *Helicobacter pylori*, and viral infections, including herpes simplex virus 1, Epstein–Barr virus, cytomegalovirus, and so on, have been reported to be related to the etiology of BD [21]. Molecular mimicry between microbes and autoantigens due to sequence homology rather than direct infection in the development of BD is considered a possible mechanism [17,22]. For example, the sequence homology between microbial and human heat-shock proteins, streptococcal cell wall M proteins and tropomyosin was reported to be associated with autoimmune reactions in BD [23,24,25].

In addition, as for intestinal BD, the location of lesions has direct contact with intestinal flora, and there may be a close relationship between gastrointestinal involvement of BD and the intestinal microbiome in theory. Several previous studies reported changes in the intestinal flora in patients with BD. By comparing the fecal microflora of BD patients with healthy controls, it was found that patients had significantly lower flora diversity and decreased short-chain fatty acid content in the intestine, which could cause intestinal epithelial barrier damage and imbalance in T-cell differentiation [26,27,28]. A Japanese study found that the relatively high abundance of *Bifidobacteria* in the intestinal flora of BD patients may trigger immune disorders by affecting intestinal pH [28]. A recent study has found deviations in microbiota composition between BD patients with skin mucosa, ocular and vascular involvement [29]. However, these studies did not separately analysis intestinal BD patients; some even excluded intestinal BD patients. There has been a lack of studies on the interaction between the development of intestinal BD and the imbalance of the intestinal microbiome thus far, which is a gap needed to be filled urgently.

### 2.2. Cellular Immunity in Adaptive Immunity

In the adaptive immunity of BD, cellular immunity is considered to play an important role, and T cells are the main lymphocytes, mainly Th1 cells, regulatory T cells (Tregs), and Th17 cells [30,31]. Studies have found mixed Th1 and Th2 cytokines in BD patients with different organ involvement [32,33,34]. Imamura et al. found that intestinal BD lesions expressed interferon (IFN)-γ, TNF-α, and interleukin (IL)-12, which were signature Th1 cytokines. Th1-related CCR5 was detected in intestinal samples from BD, while Th2-related CCR3 and CCR4 were mainly in inactive patients. This evidence of Th1 polarization indicated that Th1-dominant immune response has a close association with the pathogenesis of intestinal BD [35]. In addition, accumulating evidence has revealed that the expression levels of Th17 cells and related cytokines such as IL-17, IL-21, IL-22, and IL-23 are significantly higher in active BD than in inactive BD and BD at the remission stage [36,37,38]. In intestinal BD, the role of Th17 cells in the development of gastrointestinal involvement in BD is controversial. Ferrante et al. reported that the expression levels of Th1-related cytokines, such as TNF-α, IFN-γ, IL-12, and IL-27, were upregulated at the mRNA level in 11 patients with intestinal BD compared with healthy controls, which could inhibit the differentiation of Th0 cells into Th17 cells [30]. However, another study showed that partial CD4 clones isolated from the intestinal mucosa of eight patients at the early stage of intestinal BD produced IFN-γ and IL-17 (Th1/Th17 profile), suggesting that both Th1 and Th17 cells drive inflammation leading to mucosal damage in the early stage of intestinal BD [31]. Considering the small number of included patients, studies with a large sample size are required to explore the roles of Th1 and Th17 cells in intestinal BD. Tregs are considered a sublineage of CD4+ T cells and have a critical role in the regulation of immune tolerance and homeostasis by expressing immunosuppressive cytokines such as IL-10, IL-35, and transforming growth factor (TGF)-β as well as other functional proteins such as transcription factor forkhead box protein 3 (Foxp3) [39,40]. However, the expression levels of Tregs in BD are also inconsistent based on previous studies [41,42]. In addition, other T cells, such as Th22 cells and related cytokines, are also involved in the pathogenesis of BD. Th22 cells are CD4+ effector T cells and mainly secrete IL-22 and TNF-α, with expression of chemokine receptors as CCR4, CCR6, and CCR10. For example, the overexpression of Th22-related cytokines such as IL-22, TNF-α and chemokine receptor CCR10 was detected in the ocular samples of patients with active BD [43]. As well, the increased level of IL-22 also seemed to be associated with active uveitis and the recurrence of ulcers in patients with BD [44].

### 2.3. Humoral Immunity in Adaptive Immunity

Although cellular immunity plays a critical role in the pathogenesis of BD, humoral immunity also contributes to the immune reactions in the development of BD, as it takes effect in other autoimmune diseases [45]. For example, Suzuki et al. reported that patients with active BD had an increase in B cells spontaneously secreting immunoglobulin (IgG+IgA+IgM) and a decreased B-cell response to the T-cell-independent B-cell mitogen, such as *Staphylococcus aureus* Cowan 1. T-cell-dependent polyclonal activator, pokeweed mitogen. No response to the T-cell-dependent polyclonal activator as a pokeweed mitogen was also found in these patients, suggesting that B-cell abnormalities may be involved in the pathogenesis of BD [46]. Moreover, Eksioglu-Demiralp et al. found that compared with healthy controls, there was a remarkable increase in subsets of B cells marked as CD13, CD33, CD80, and CD45RO in patients with BD, although the total number of B cells marked as CD19+ was normal [47].

### 2.4. Innate Immunity

Innate immune cells are mainly composed of neutrophils, mononuclear phagocytes, dendritic cells, natural killer (NK) cells, γδ T cells, mast cells, and so on, of which abnormal activation plays an important role in the pathogenesis of BD [48].

Neutrophils, which act as the first line of defense against pathogens, can cause bodily injury and further the immune response, especially in patients with autoinflammatory and autoimmune disorders [49]. Previous studies reported that hyperactive neutrophils were associated with perivascular infiltration in lesion sites of patients with BD. Moreover, by increasing chemotaxis, phagocytosis, and superoxide production, neutrophils could contribute to high levels of proinflammatory cytokines such as IL-8, IL-12, TNF-α, INF-γ, and vascular endothelial dysfunction [50,51,52]. It is noteworthy that neutrophils act as a key factor in mediating thrombosis in BD [22,53].

NK cells are another important component of innate immunity, which regulates other immune cells, such as T cells and dendritic cells, and defends against infected cells and tumor cells by cytotoxicity and cytokine secretion, with IFN-γ as the hallmark cytokine [54,55,56]. Accumulating evidence has reported that the number of NK cells in the peripheral blood of patients with BD is less than that in healthy controls, suggesting that NK cells play a protective role in BD [54,57].

γδ T cells are also involved in the regulation of the autoimmune response, of which the number could increase from minority to majority of all circulating T cells in a short time once infected [58]. Sutton et al. reported that γδ T cells activated by IL-1β and IL-23 could produce IL-17, IL-21 and IL-22 without T-cell receptor engagement and mediate autoimmune inflammation in autoimmune diseases [59]. In patients with active BD, Parlakgul et al. found functional changes in γδ T cells and decreased related cytokine responses, although there was no increase in the number of γδ T cells [60].

For intestinal BD, Ahn et al. revealed that extracellular high-mobility group Box 1 (HMGB1) expression, which could activate the release of cytokines and mediate inflammation in innate immunity, was significantly increased in BD patients with gastrointestinal involvement compared to BD patients without gastrointestinal involvement and healthy controls [61]. Kirino et al. reported that Toll-like receptor 4 (TLR4), which mediates activation of the innate immune system, was upregulated with a reduction in the anti-inflammatory enzyme heme oxygenase (HO)-1 in peripheral blood mononuclear cells in patients with BD, suggesting the involvement of innate immunity in the pathogenesis of BD [62].

## 3. Present and Potential Biological Agents

The key points of intestinal BD treatment focus on the suppression of inflammatory exacerbations and the prevention of recurrences. As for intestinal BD, mucosal healing tends to have a better prognosis after the remission of clinical symptoms and normalization of C-reactive protein (CRP) [63]. Biological agents are an emerging category of pharmaceuticals for various diseases, including inflammatory diseases and cancers, in recent decades. They are defined by their derivation from biological sources, including monoclonal antibodies, fusion receptor proteins, hormones, and cytokines [64,65]. Table 2 lists the immune-targeted biological agents and possible applicable subtypes in BD according to current studies. And based on the immune mechanism of BD, including intestinal BD, biological agents are promising choices for the achievement of intestinal BD treatment targets. We list present and potential biological agents in intestinal BD as well as related targets, including pathogenic cells, cytokines, and pathways.

### 3.1. Anti-TNF-α Agents

TNF-α is an important proinflammatory cytokine in both innate immunity and adaptive immunity in BD patients with significantly elevated serum concentrations and mediating mucosal damage [103]. TNF-α antagonists can be classified into two groups based on different mechanisms of action: a. monoclonal antibodies such as infliximab, adalimumab, and golimumab. b. soluble receptors such as etanercept. Anti-TNF-α agents such as infliximab and adalimumab have been proven to be rapidly effective in inducing and maintaining remission of intestinal BD and are recommended by various international guidelines of intestinal BD as the only definitely effective biological agent [16,104,105]. Tanida et al. conducted a study at 12 sites in Japan in patients with intestinal BD (*n* = 20) who were refractory to corticosteroid and/or immunomodulator therapies. The results showed that most patients achieved improvement in global gastrointestinal symptoms and endoscopic scores at weeks 24 and 52, with complete remission in 20% of patients at weeks 24 and 52. No new safety signals were observed and no death occurred. Four of six patients experienced an infection after the dose escalated [106]. Another prospective study conducted by Zou et al. in patients with moderate-to-severe active intestinal BD (*n* = 27) revealed that 84.6%, 70%, and 70% of patients achieved clinical responses at weeks 14, 30, and 52 after infliximab treatment, respectively, with proportions of clinical remission of 69.2%, 40%, and 55%. Infliximab therapy was generally well tolerated in all patients, and five patients (18.5%) developed infectious adverse events [107]. Recently, Zhang et al. performed a systemic and meta-analysis including 13 studies on the application of anti-TNF-α agents in patients (*n* = 739) with intestinal BD, suggesting that anti-TNF agents, including infliximab and adalimumab, were an efficient therapy for intestinal BD. The pooled clinical remission rate at Months 3, 6, 12, and 24 were 0.61 (95%CI 0.48–0.78), 0.51 (95%CI 0.40–0.66), 0.57 (95%CI 0.48–0.67), and 0.38 (95%CI 0.16–0.88), respectively. The pooled mucosal healing rate at Months 3, 6, 12, and 24 were 0.66 (95%CI 0.50–0.86), 0.82 (95%CI 0.48–0.98), 0.65 (95%CI 0.51–0.81), and 0.69 (95%CI 0.39–1.00), respectively. The pooled estimate of the proportion of overall adverse reactions for infliximab was 0.22 (95%CI 7–69%). The majority was acute or delayed infusion reaction or infection. The pooled proportions of infusion reactions and infection were respectively 12% (95%CI 5–29%) and 21% (95%CI 6–80%) related to infliximab [108].

### 3.2. IFN-α

IFN-α has antiviral and antitumor functions as well as involvement in the regulation of the cytokine network and innate and adaptive immunity [109]. Because of the hypothesis of viral pathogenesis, treatment with IFNs has been suggested [73]. Data reporting the efficacy of IFN-α in BD patients can be traced back to 1986 [110]. Accumulating evidence has proven its positive role in BD, especially with ocular involvement [111,112,113,114,115,116,117,118,119,120]. IFNα may prompt the reperfusion of occluded vessels, resulting in complete remission of ocular vasculitis [121]. The EULAR updated the recommendations for BD in 2018 and recommended the application of IFN-α in BD patients with mucocutaneous, articular, ocular, and vascular involvement [5]. However, there is a lack of data about the efficacy of IFN-α in intestinal BD, except for a few case reports. Monastirli et al. reported a 40-year-old woman with BD under severe conditions with acute myelitis and intestinal involvement who achieved a nearly complete remission at week 10 after treatment with IFN-α and conventional medications. In this case report, complete remission of intestinal ulcers was observed after a 9-day combined application of IFN-α with corticosteroids followed by a 4-week IFN-a monotherapy [72]. Another two case reports showed that patients with ocular and gastrointestinal involvement at the same time experienced the disappearance of peptic ulcers after treatment with IFN-α-targeting ophthalmitis [73,74]. Based on these limited experiences, potential side effects mainly include flu-like syndrome (such as fever and arthralgia) and transient leukopenia. Further large-sample studies are required to validate the efficacy and safety of IFN-α treatment in intestinal BD.

### 3.3. IL-1 Antagonist

Previous studies have revealed that IL-1 acts as an important proinflammatory cytokine in the development of inflammatory diseases by inducing an acute phase response, activating endothelial cells, and expressing cell adhesion molecules and coagulation factors [122]. Elevated IL-1 levels have been found in BD since 1990 and have been confirmed in subsequent studies, laying a theoretical foundation for the use of an IL-1 antagonist as a therapeutic agent in BD [123,124]. IL-1 antagonists inhibit local inflammatory effects of IL-1, mainly including IL-1 receptor antagonists such as anakinra (ANA) and anti-IL-1β humanized monoclonal antibodies such as canakinumab (CAN) and gevokizumab (GEV). The efficacy of ANA and CAN in patients with BD was mainly reported in retrospective case series, which were not specific to intestinal BD [77,78,79]. Cantarini et al. conducted a study in BD patients (*n* = 9) with multiorgan involvement who were refractory to anti-TNF-α agents and standardized therapies. A total of 33.3% of patients had gastrointestinal involvement. Within 1 or 2 weeks after treatment with ANA, eight out of nine patients received a prompt response, while most patients experienced a relapse over time for an unknown reason. With regard to anakinra safety, three of nine BD patients suffered a mild itchy skin rash at the sites of anakinra injection. No serious adverse events occurred in all nine patients [75]. Another study conducted by Vitale et al. found that three adult patients with BD who were refractory to conventional medications all achieved prompt and sustained clinical remission after treatment with CAN, of whom two had gastrointestinal involvement [76]. GEV did not obtain a significantly positive result in a phase 3 trial, although a rapid clinical response and inflammation control were observed in BD patients refractory to conventional treatments in a phase 2 trial [80,81]. Certainly, future large-sample studies are needed.

### 3.4. IL-6 Antagonist

IL-6 is another important proinflammatory cytokine in the innate immunity of BD that is usually produced by NK cells and γδ T cells [125,126]. IL-6 usually increases TH2 differentiation and inhibits TH1 differentiation. Under inflammatory conditions, IL-6, accompanied by IL-1β and IL-21, plays a key role in the relationship between Treg and Th17 cells by inducing the differentiation of naive T cells into Th17 cells and promoting the expansion of differentiated Th17 cells [126,127,128]. Both Th1/Th2 and Th1/Th17 profiles have been discussed in the pathogenesis of BD, providing support for the use of IL-6 antagonists. Tocilizumab is a humanized monoclonal antibody that can block IL-6-mediated proinflammatory reactions by specific binding of the IL-6 receptor [129]. Accumulating evidence based on small-scale studies revealed that tocilizumab was effective for refractory ocular, neurological and vascular BD, while it did not play a positive role in BD patients with gastrointestinal, mucocutaneous, and articular involvement [83,84,130]. In contrast, there is one case report presenting a positive effect of tocilizumab on intestinal BD. Chen et al. reported in 2016 that a young female patient with intestinal BD was refractory to multiple conventional medications with limited application of anti-TNF-α agents due to side effects. Therefore, tocilizumab was considered a therapeutic option. Symptoms and endoscopic images of this patient improved during nine months of administration and no adverse events were reported [85]. Undoubtedly, further studies are required for a definite conclusion.

### 3.5. IL-17 Antagonist

As mentioned in the Immunity mechanism section in our review, IL-17 also plays a proinflammatory role in the pathogenesis of BD. Previous studies revealed that serum IL-17 concentration and TH17 cells and the number of Th17 cells were higher in active BD than in inactive BD, which also showed a positive relationship with other inflammatory markers, such as erythrocyte sedimentation rate (ESR) and CRP [38,131]. Secukinumab, which has been approved for the treatment of psoriatic arthritis and ankylosing spondylitis, is a human IL-17A-binding monoclonal antibody [132]. However, its role in intestinal BD is controversial and needs further investigation. Di Scala G et al. conducted a study in five BD patients with mucocutaneous and joint involvement who were refractory to at least one anti-TNF-α agent. The results revealed that all patients could achieve complete response after treatment with secukinumab at a dose of 300 mg/month. The treatment was well tolerated and no significant drug reaction was observed. Only mild infections of the urinary tract were recorded in two patients [86]. Fagni et al. conducted a multicenter retrospective study of BD patients (*n* = 15) with mucosal and articular subphenotypes who were refractory to conventional medications and at least one anti-TNF-α agent. After treatment with secukinumab, the proportion of patients achieving a response (complete or partial) increased with follow-up time: 66.7% at 3 months, 86.7% at 6 months, 76.9% at 12 months, 90.0% at 18 months and 100.0% after 24 months. No serious or dose-related adverse effects were observed [87]. In contrast, in a randomized, controlled clinical trial of 118 patients with Behçet’s uveitis, there were no statistically significant differences in the primary outcome of uveitis recurrence between the secukinumab treatment groups and placebo groups, although treatment with secukinumab may reduce the use of concomitant immunosuppressive medication based on the secondary outcome of the randomized controlled trial (RCT). Greater percentages of patients in the secukinumab treatment groups experienced ocular or nonocular adverse events compared with the placebo group [88]. Barrado-Solís et al. also reported that BD developed in two patients with psoriasis a few weeks after receiving secukinumab treatment [133]. Although no related studies have been conducted in intestinal BD thus far, it is worth noting that in a high-quality RCT of 59 patients with moderate to severe CD, treatment with secukinumab has been proven to be ineffective and to have higher rates of adverse events than placebo. Drug-related serious adverse events, including worsening of CD, pilonidal cyst, and ileostomy, occurred in six patients receiving secukinumab. Twenty infections were seen on secukinumab while none on placebo [134]. Considering that intestinal BD is similar to CD in its clinical symptoms, imaging findings and endoscopic characteristics, exploration of secukinumab treatment in intestinal BD should be more rigorous and careful.

### 3.6. IL-12/IL-23 Antagonist

IL-12 and IL-23, which both belong to the IL-12 cytokine family, share the same p40 subunit [135,136]. The former has a significant role in differentiating naive T cells into Th1 cells, and the latter is indispensable for the differentiation of Th17 cells [137]. Previous studies have shown that IL-12 and IL-23 are involved in the immune mechanism of BD [138,139]. Genome Wide Association Studies (GWAS) have reported that IL12R-IL23RB2 region is associated with BD, suggesting the role of IL12 and IL23 receptors in the disease onset [140]. Ustekinumab, a fully humanized monoclonal antibody targeting the p40 subunit of IL-12 and IL-23, has been reported to be effective in BD, although there are no related studies in intestinal BD [91,92,93]. In addition, ustekinumab has been proven to be effective in treating active CD. The rates of adverse events were similar in ustekinumab and placebo group, mainly including infections, arthralgia, headache, and nausea [89,90]. In summary, considering the similarity between CD and intestinal BD, ustekinumab may be a potential biological agent in intestinal BD.

### 3.7. Small Molecule Targeted Agents

Small molecule targeted agents are chemically synthesized but have high selectivity and efficacy similar to those of typical biological agents, which were considered special biological agents in a broad sense. Among them, baricitinib as an oral Janus kinase (JAK)-inhibitors (JAKi) and apremilast as an oral phosphodiesterase 4 inhibitor, were reported in case series to be potential drugs in the treatment of intestinal BD [94,95]. Baricitinib, a new class of targeted synthetic DMARDs (tsDMARDs), interferes with signal transduction pathways of a variety of cytokines. It can suppress the differentiation of plasmablasts, Th1 and Th17 cells, as well as innate immunity, showing potential in immune-related diseases [141]. Thirteen intestinal BD patients received baricitinib 2–4 mg daily, with background glucocorticoids and immunosuppressants. 76.92% (10/13) patients achieved complete remission of global gastrointestinal symptom scores, and 66.7% (6/9) had mucosal healing on endoscopy. The disease activity index for intestinal Behçet’s disease (DAIBD) and CRP level decreased significantly [94]. Certainly, future large-sample studies are required to confirm the potential of these biological agents.

### 3.8. Other Biological Agents

In addition to the drugs mentioned above, there are other biological agents with potential applications in intestinal BD as well. Considering the lack of related studies in intestinal BD, our review plan is to perform a preliminary overview of these drugs: a. Rituximab, a chimeric mouse/human monoclonal antibody, could cause B-cell lysis by specifically binding CD20 antigen on the B lymphocyte. Previous studies have shown that it may have positive effects on BD with mucocutaneous, articular, neurological, and ocular subphenotypes. According to the pilot RCT study, in the rituximab group, two of ten patients had conjunctivitis during the first infusion, one patient had pneumonia 4 months after the infusion and one patient had herpes zoster 4 months after the infusion. The remaining side effects were infusion-related, all during the first infusion [97,98]. b. Abatacept, a soluble fusion protein consisting of the extracellular domain of human cytotoxic T-lymphocyte-associated protein 4 (CTLA4), is a selective T-cell costimulation modulator and a protein drug for autoimmune diseases [142]. A case report published by Maciel suggested that abatacept may be effective for refractory ocular and mucocutaneous BD [99]. c. Alemtuzumab, a humanized monoclonal antibody against CD52, is a glycoprotein expressed on the surface of most lymphocytes and in myeloid cells to a lesser extent. Therefore, it could induce deep depletion of lymphocytes and then restore T cells and B cells with a regulatory phenotype [143]. Previous studies revealed that alemtuzumab may provide an alternative treatment strategy for refractory BD with ocular, vascular, and neurological involvement [100,101,144]. d. Vedolizumab, a humanized anti-α4β7 integrin monoclonal antibody, can modulate gut lymphocyte trafficking. It has been approved for both induction and maintenance treatment of inflammatory bowel diseases (IBD) such as CD and ulcerative colitis. Its gastrointestinal-specific interaction reduces the side effects associated with systemic immunosuppression [90]. Up to now, there has been only one case report of a patient with intestinal BD suggesting its efficacy as a valid option for treatment. This 49-year-old female was unsuccessfully treated with conventional immunosuppressive and several biological agents and received vedolizumab at 0, 2, and 6 weeks and then every 4 weeks. After the second dose of vedolizumab, a marked improvement of intestinal BD was achieved and clinical remission was achieved at 6 months without side effects [102].

## 4. Conclusions

Intestinal BD has been classified into IBD to some extent, of which typical members are UC and CD, considering the similarity of genetic backgrounds, clinical manifestations, and therapeutic strategies. As well, there is no doubt that progress in our understanding of the pathogenesis of intestinal BD as well as the other members of IBD will pave the way for seeking effective approaches. Based on the immunological perspective, we have come to know that immune cells, related cytokines, and specific autoantibodies play important roles in the pathogenesis of intestinal BD, although the etiology remains unclear. Biological agents targeting potential pathogenic cells, cytokines, and pathways are optimized options in the treatment of intestinal BD. Some of them have been considered to be definitely effective in intestinal BD and are widely used in clinical practice, such as anti-TNF-α agents, while others require further well-designed multi-center RCT or other study designs that would provide high-quality evidence to confirm their efficacy. Detailed issues accompanied by the wide use of biological agents in the near future, such as adjustment of dosage form or dosage for an invalid biological agent, transition to other biological agents, combination of different kinds of drugs and prevention of adverse effects, call for in-depth thinking and further extensive studies. Certainly, drugs related to immune mechanisms in preclinical stages also deserve attention. Furthermore, considering that intestinal BD is a specific and less common phenotype of BD, separate studies focused on intestinal involvement and comparisons of different phenotypes will become the trend of future research, especially from the perspective of pathogenesis. These will provide more precise explanations of mechanisms and more individualized treatment strategies for different organ involvement.

## Figures and Tables

**Table 1 ijms-24-08176-t001:** The clinical characteristics of intestinal BD and CD.

	Intestinal Behcet’s Disease	Crohn’s Disease
Lesion distribution	Common in ileocecal region, rare in rectum and anus, short segment lesions	Common in ileocecal region, long segment lesions, jumping distribution
Bowel morphology	Not prone to stenosis	Thickening and stenosis
Gastrointestinal manifestations	Abdominal pain, diarrhea, hematochezia, with or without abdominal mass sometimes	Abdominal pain, diarrhea, hematochezia, abdominal mass, with or without perianal lesion
Extra-gastrointestinal manifestations	Oral and vulval ulcers, folliculitis or acne-like skin lesions, systemic manifestations (for example, ocular, vascular, neurological and articular symptoms)	Oral ulcers, nodular erythema, pyoderma, arthritis and so on
Laboratory tests	Positive in acupuncture test, HLA-B5 and ASCA	Positive in ASCA
Endoscopic findings	Round or oval ulcers, volcano-like ulcers, single or multiple ulcers ≤ 5, with definite boundary and smooth mucosa around the ulcer	Discontinuous distribution of longitudinal ulcers, paving stone-like pattern, aphthous ulcers
Pathologic findings	Signs of vasculitis.	Transmural inflammation, fissure-like ulcers, non-caseous granuloma

BD: Behcet’s disease; CD: Crohn’s disease; HLA-B5: human leukocyte antigen⁃B5; ASCA: anti⁃saccharomyces cerevisiae antibody.

**Table 2 ijms-24-08176-t002:** Immune-targeted biological agents and possible applicable subtypes in BD.

Biological Agents	Immune-Related Targets	Structure	Possible Applicable Subtypes in BD
Infliximab, adalimumab	TNF-α	Monoclonal antibodies against TNF-α	All subtypes of BD * [5]
Golimumab			Intestinal BD [66]; BD with ocular and neurological involvement [67,68,69,70]
Etanercept		Soluble receptors against TNF-α	Intestinal BD [71]; BD with mucocutaneous and articular involvement * [5]
IFN-α	Not clear	Recombinant human IFN-α-2a	Intestinal BD [72,73,74]; BD with mucocutaneous, articular, ocular, and vascular involvement * [5]
Anakinra	IL-1	Recombinant human IL-1 receptor antagonist	Intestinal BD [75,76]; BD with mucocutaneous * and ocular involvement [77,78,79]
Canakinumab		Anti-IL-1β humanized monoclonal antibodies	
Gevokizumab			Controversial in intestinal BD [80,81]; BD with ocular involvement [80,82]
Tocilizumab	IL-6	Human IL-6 receptor monoclonal antibody	Controversial in intestinal BD; BD with ocular, neurological, and vascular involvement [83,84,85]
Secukinumab	IL-17	Human IL-17A monoclonal antibody	Unclear in intestinal BD; BD with mucocutaneous and articular involvement [86,87,88]
Ustekinumab	IL-12/IL-23	Human IL-12/IL-23p40 monoclonal antibody	Unclear in intestinal BD but effective in CD [89,90]; BD with mucocutaneous and ocular involvement [91,92,93]
Baricitinib	JAK1/JAK2	JAK1/JAK2 inhibitor; small molecule drug	Intestinal BD [94]
Apremilast	Phosphodiesterase 4	Phosphodiesterase 4 inhibitor; small molecule drug	Intestinal BD [95]; BD with mucocutaneous [95,96]
Rituximab	CD20	Chimeric mouse/human monoclonal antibody against CD20 antigen on the B lymphocyte	Unclear in intestinal BD; BD with mucocutaneous, articular, neurological, and ocular involvement [97,98]
Abatacept	B7	Selective T-cell costimulation modulator and a protein drug	Unclear in intestinal BD; BD with mucocutaneous and ocular involvement [99]
Alemtuzumab	CD52	Humanized monoclonal antibody against CD52	Unclear in intestinal BD; BD with ocular, vascular, and neurological involvement [100,101]
Vedolizumab	α4β7 integrin	Humanized anti-α4β7 integrin monoclonal antibody	Intestinal BD [102]

BD: Behcet’s disease; *: Recommendation from international guidelines; IL: interleukin; CD: Crohn’s disease; JAK: Janus kinase.

## Data Availability

The data underlying this article are available in the article.

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
