# Peer review of "Intestinal Behcet’s Disease: A Review of the Immune Mechanism and Present and Potential Biological Agents"

_ijms, 2023, doi:10.3390/ijms24098176_

Round 1
Reviewer 1 Report
The paper discusses the chronic and recurrent systemic vasculitis known as Behcet's Disease (BD) focusing on the biological agents eligible as a therapeutic strategy. The authors delve into the dysregulation of immune function as one of the main pathogeneses and highlight the emergence of biological agents as an effective treatment option for patients with refractory intestinal BD. Overall, the review is well-written and well-organized.
A few concerns are suggested to be addressed as follows:
- Lines 96-98: " of which IFN-γ and IL-12 are signature cytokines, are associated with the activity of BD, suggesting that Th1 immune reactions are critical in the pathogenesis of BD[29-31]" Authors should better describe this point.
- Lines 123-124: Which immunoglobulin are the authors referring to?
Author Response
- - Lines 96-98: " of which IFN-γ and IL-12 are signature cytokines, are associated with the activity of BD, suggesting that Th1 immune reactions are critical in the pathogenesis of BD[29-31]" Authors should better describe this point.
Thank you for your summary and comments. We have provided a further explanation about the role of Th1 immune reactions in the pathogenesis of BD in the revised version (Lines 144-151) which are as follows:
In the adaptive immunity of BD, cellular immunity is considered to play an important role, and T cells are the main lymphocytes, mainly Th1 cells, regulatory T cells (Tregs), and Th17 cells[30, 31]. Studies have found mixed Th1 and Th2 cytokines in BD patients with different organ involvement [32-34]. Imamura et al found that intestinal BD lesions expressed interferon (IFN)- γ, tumor necrosis factor (TNF)- α and interlukin (IL)-12, which were signature Th1 cytokines. Th1 related CCR5 were detected in intestinal samples from BD, while Th2-related CCR3 and CCR4 were mainly in inactive patients. These evidences of Th1 polarization indicated that Th1-dominant immune response has close association with the pathogenesis of intestinal BD[35]
- - Lines 123-124: Which immunoglobulin are the authors referring to?
Response: Thank you for comments. The spontaneously secreting immunoglobulin in Lines 123-124 was the summation of IgG, IgA and IgM according to the original study. And in the revised version (Lines 250-251), the additional notes are added in the corresponding section of the latest version of our manuscript which are as follows:
For example, Suzuki et al. reported that patients with active BD had an increase in B cells spontaneously secreting immunoglobulin (IgG+IgA+IgM) and a decreased B-cell response to the T-cell-independent B-cell mitogen, such as Staphylococcus aureus Cowan 1.
Reviewer 2 Report
"Intestinal Behcet's Disease: A Review of the Immune Mechanism and Present and Potential Biological Agents" is a comprehensive and well-written article that provides an in-depth overview of the immune mechanism underlying intestinal Behcet's disease (BD) and highlights the potential of biological agents as treatment options. The authors have successfully synthesized the current understanding of this complex disease and its immune pathogenesis, making it a valuable resource for researchers, clinicians, and healthcare professionals.
One of the strengths of this article is the thorough review of the immune mechanism of intestinal BD. The authors have clearly outlined the dysregulation of immune function as one of the main pathogeneses of the disease, although the exact etiology remains unclear. They have provided a detailed overview of the role of immune cells, cytokines, and autoantibodies in the pathogenesis of intestinal BD, which is crucial for understanding the disease process.
The article also highlights the emergence of biological agents as a promising therapeutic approach for intestinal BD. The authors have meticulously reviewed the literature on the use of biological agents, particularly anti-tumor necrosis factor (TNF) agents, in the treatment of refractory intestinal BD. They have emphasized the effectiveness of these agents in patients who show poor response to conventional medications and are at risk of surgery. The review provides valuable insights into the optimized options of biological agents that target potential pathogenic cells, cytokines, and pathways, based on the deep understanding of the immune mechanism of intestinal BD. This information is highly relevant for clinicians and researchers involved in the management of patients with intestinal BD, as it highlights the potential of these agents in improving patient outcomes.
Furthermore, the authors have acknowledged the need for further research to confirm the efficacy of other biological agents in the treatment of intestinal BD. They have rightly pointed out that large-sample and high-quality studies are necessary to establish the effectiveness of these agents. They have also highlighted important practical issues associated with the use of biological agents, such as dosage adjustments, transition to other agents, combination therapies, and prevention of adverse effects. This demonstrates the authors' comprehensive approach towards addressing the practical implications of using biological agents in the management of intestinal BD, and their recognition of the challenges and complexities involved in real-world clinical practice.
The article is well-structured and provides a clear and concise overview of the immune mechanism and potential biological agents for intestinal BD. The language used is appropriate for a scientific article, and the authors have provided ample references to support their statements and findings. The inclusion of keywords in the abstract and the conclusion section is also helpful in summarizing the key points of the article and making it easily searchable.
However, there are a few areas that could be further improved. Firstly, while the article provides a comprehensive overview of the immune mechanism and potential biological agents for intestinal BD, it would be beneficial to include more specific details on the mechanisms of action of these biological agents, as well as their efficacy and safety profiles. This would provide a more in-depth understanding of the therapeutic potential of these agents in the management of intestinal BD.
Secondly, while the authors have acknowledged the need for further research to confirm the efficacy of other biological agents, it would be helpful to provide suggestions for future research directions in this area. This could include recommendations for well-designed randomized controlled trials or other study designs that would provide high-quality evidence for the effectiveness of these agents in intestinal BD.
Despite these minor limitations, "Intestinal Behcet's Disease: A Review of the Immune Mechanism and Present and Potential Biological Agents" is a well-written and informative article that contributes to the current knowledge of the immune pathogenesis of intestinal BD and the potential of biological agents as treatment options.
Author Response
However, there are a few areas that could be further improved.
Firstly, while the article provides a comprehensive overview of the immune mechanism and potential biological agents for intestinal BD, it would be beneficial to include more specific details on the mechanisms of action of these biological agents, as well as their efficacy and safety profiles. This would provide a more in-depth understanding of the therapeutic potential of these agents in the management of intestinal BD.
Response: We really appreciate your efforts in reviewing our manuscript. We have provided more details about the mechanisms, efficacy and safety of certain biological agents in our review, including IFN-α (Part 3.2), IL‐1 antagonist (Part 3.3), IL‐6 antagonist (Part 3.4), IL-12/IL-23 antagonist (Part 3.6), and JAKi (Part 3.7).
Secondly, while the authors have acknowledged the need for further research to confirm the efficacy of other biological agents, it would be helpful to provide suggestions for future research directions in this area. This could include recommendations for well-designed randomized controlled trials or other study designs that would provide high-quality evidence for the effectiveness of these agents in intestinal BD.
Response: Thanks for you suggestions. We have added our recommendations for further studies in the conclusions part, including more high-quality RCT about treatment and more attention to the subtype of intestinal BD in the part of ‘Conclusions’ which are as follows:
Biological agents targeting potential pathogenic cells, cytokines and pathways are optimized options in the treatment of intestinal BD. Some of them have been considered to be definitely effective in intestinal BD and are widely used in clinical practice, such as anti-TNF-α agents, while others require further well-designed multi-center RCT or other study designs that would provide high-quality evidence to confirm their efficacy. Detailed issues accompanied by the wide use of biological agents in the near future, such as adjustment of dosage form or dosage for an invalid biological agent, transition to other biological agents, combination of different kinds of drugs and prevention of adverse effects, call for in-depth thinking and further extensive studies. Furthermore, considering that intestinal BD is a specific and less common phenotype of BD, separate studies focused on intestinal involvement and comparisons of different phenotypes will become the trend of future research, especially from the perspective of pathogenesis. These will provide more precise explanations of mechanisms and more individualized treatment strategies for different organ involvement.
Reviewer 3 Report
The article's subject matter is appropriate for the journal's audience, and the author has pointed out the ways in which the most recent findings may advance the discipline. In my opinion, the text as a whole is quite well written; each chapter is well-organized and flows well into the next. The research has a clear and accurate title. The abstract provides a succinct overview of the whole project. The Introduction is well-structured, giving the reader the necessary context for the rest of the paper. The merits of the research are discussed enough in the last section of the paper (the conclusion). However, some minor questions authors need to be addressed.
1. The author covered the small molecules and other biological agents which are all standard products. I suggest adding some molecules or agents which are in the clinical or preclinical stage.
2. It would be better if the authors added tables of clinically studied compound data and pictorial presentation of mechanisms.
Author Response
- The author covered the small molecules and other biological agents which are all standard products. I suggest adding some molecules or agents which are in the clinical or preclinical stage.
Response: We really appreciate your suggestions and will devote continuous attention to potential molecules and biological agents in the preclinical stage in the future. And lots of molecules and biological agents mentioned in our articles are potential drugs reported in clinical studies based on related immune mechanism, which are not recommended in the international guidelines, calling for further high-quality studies to confirm efficacy. We have added what you suggested in the part of ‘Conlucions’ which are as follows:
Biological agents targeting potential pathogenic cells, cytokines and pathways are optimized options in the treatment of intestinal BD. Some of them have been considered to be definitely effective in intestinal BD and are widely used in clinical practice, such as anti-TNF-α agents, while others require further well-designed multi-center RCT or other study designs that would provide high-quality evidence to confirm their efficacy. Detailed issues accompanied by the wide use of biological agents in the near future, such as adjustment of dosage form or dosage for an invalid biological agent, transition to other biological agents, combination of different kinds of drugs and prevention of adverse effects, call for in-depth thinking and further extensive studies. Certainly, drugs related to immune mechanisms in preclinical stages also deserve attention. Furthermore, considering that intestinal BD is a specific and less common phenotype of BD, separate studies focused on intestinal involvement and comparisons of different phenotypes will become the trend of future research, especially from the perspective of pathogenesis. These will provide more precise explanations of mechanisms and more individualized treatment strategies for different organ involvement.
- It would be better if the authors added tables of clinically studied compound data and pictorial presentation of mechanisms.
Response: Thanks for your suggestions, we have added two tables in the revised version.
Reviewer 4 Report
This article provides key information on the immunological mechanism of the disease and possible biological therapies. The authors focus on the serious complications associated with intestinal involvement, such as haemorrhage, perforation and obstruction, as well as the growing role of biological therapies in the treatment of patients who do not respond to conventional treatment.
The authors cite a number of articles in the paper that, among other things, point to the important role of environmental factors in genetically predisposed patients. Among these factors are viral and bacterial infections, which, the authors note, can affect changes in the gut flora. It is worth mentioning that there are numerous studies not cited by the authors that support this claim.
In the remainder of the article, the authors address issues related to biological agents, which are currently an important topic in the gastroenterology community. The paper presents various methods and attempts to treat intestinal diseases, including the use of biologic agents. All cases cited are based on current data and research.
The article has been prepared with the utmost care and the selection of literature cited is reliable and relevant to the topic presented. It is also worth noting that the dates of the cited articles are up-to-date, which further emphasises the factual value of this article.
Author Response
The authors cite a number of articles in the paper that, among other things, point to the important role of environmental factors in genetically predisposed patients. Among these factors are viral and bacterial infections, which, the authors note, can affect changes in the gut flora. It is worth mentioning that there are numerous studies not cited by the authors that support this claim.
Response: We appreciate the reviewer’s suggestions. We have supplied more information about the important role of intestinal microbiota in BD pathogenesis (Part 2.1. Immunogens). Certainly, as we mainly focused on the immune mechanism in the pathogenesis, we cited several studies to explain the relationship between gut flora and immune mechanism in the intestinal BD which are as follows:
In addition, as for intestinal BD, the location of lesions has direct contact with intestinal flora, and there may be a close relationship between gastrointestinal involvement of BD and the intestinal microbiome in theory. Several previous studies reported changes in the intestinal flora in patients with BD. By comparing the fecal microflora of BD patients with healthy controls, it was found that patients had significantly lower flora diversity and decreased short-chain fatty acid content in the intestine, which could cause intestinal epithelial barrier damage and imbalance in T-cell differentiation[26-28]. A Japanese study found that the relatively high abundance of bifidobacteria in the intestinal flora of BD patients may trigger immune disorders by affecting intestinal pH[28]. A recent study has found deviations in microbiota composition between BD patients with skin mucosa, ocular and vascular involvement[29]. However, these studies did not separately analysis intestinal BD patients, some even excluded intestinal BD patients. There has been a lack of studies on the interaction between the development of intestinal BD and imbalance of the intestinal microbiome thus far, which is a gap needed to be filled urgently.
References:
[26] Consolandi C, Turroni S, Emmi G, Severgnini M, Fiori J, Peano C, et al. Behçet's syndrome patients exhibit specific microbiome signature. Autoimmun Rev. 2015; 14(4):269-276. doi: 10.1016/j.autrev.2014.11.009.
[27] Shimizu J, Kubota T, Takada E, Takai K, Fujiwara N, Arimitsu N, et al. Relative abundance of Megamonas hypermegale and Butyrivibrio species decreased in the intestine and its possible association with the T cell aberration by metabolite alteration in patients with Behcet's disease (210 characters). Clin Rheumatol. 2019; 38(5):1437-1445. doi: 10.1007/s10067-018-04419-8.
[28] Shimizu J, Kubota T, Takada E, Takai K, Fujiwara N, Arimitsu N, et al. Bifidobacteria Abundance-Featured Gut Microbiota Compositional Change in Patients with Behcet's Disease. PLoS One. 2016; 11(4):e0153746. doi: 10.1371/journal.pone.0153746.